# Mechanisms of Glucose Absorption in the Small Intestine in Health and Metabolic Diseases and Their Role in Appetite Regulation

**DOI:** 10.3390/nu13072474

**Published:** 2021-07-20

**Authors:** Lyudmila V. Gromova, Serguei O. Fetissov, Andrey A. Gruzdkov

**Affiliations:** 1Pavlov Institute of Physiology, Russian Academy of Sciences, 199034 Saint-Petersburg, Russia; Serguei.Fetissov@univ-rouen.fr (S.O.F.); gruzdkovaa@infran.ru (A.A.G.); 2Neuronal and Neuroendocrine Differentiation and Communication Laboratory, Inserm UMR1239, University of Rouen Normandy, 76130 Mont-Saint-Aignan, France; 3Pavlov’s Department of Physiology, Institute of Experimental Medicine, 197376 Saint-Petersburg, Russia

**Keywords:** glucose, small intestine, insulin resistance, gut-brain axis, appetite, obesity metabolic syndrome, type 2 diabetes mellitus

## Abstract

The worldwide prevalence of metabolic diseases such as obesity, metabolic syndrome and type 2 diabetes shows an upward trend in recent decades. A characteristic feature of these diseases is hyperglycemia which can be associated with hyperphagia. Absorption of glucose in the small intestine physiologically contributes to the regulation of blood glucose levels, and hence, appears as a putative target for treatment of hyperglycemia. In fact, recent progress in understanding the molecular and cellular mechanisms of glucose absorption in the gut and its reabsorption in the kidney helped to develop a new strategy of diabetes treatment. Changes in blood glucose levels are also involved in regulation of appetite, suggesting that glucose absorption may be relevant to hyperphagia in metabolic diseases. In this review we discuss the mechanisms of glucose absorption in the small intestine in physiological conditions and their alterations in metabolic diseases as well as their relevance to the regulation of appetite. The key role of SGLT1 transporter in intestinal glucose absorption in both physiological conditions and in diabetes was clearly established. We conclude that although inhibition of small intestinal glucose absorption represents a valuable target for the treatment of hyperglycemia, it is not always suitable for the treatment of hyperphagia. In fact, independent regulation of glucose absorption and appetite requires a more complex approach for the treatment of metabolic diseases.

## 1. Introduction

Glucose is a major macronutrient and a vital homeostatic factor in the regulation of energy metabolism maintained in a narrow range of 4.4 to 6.1 mmol/L or about 1.0 g/L in the blood of healthy humans as measured in the fasting state. However, glucose per se is not the predominant component of mixed food, and its main source in the diet is poly- and oligosaccharides, which undergo enzymatic hydrolysis to monomers in the small intestine during luminal and membrane digestion [1]. Depending on the food composition, the site of the gastrointestinal tract (GIT) and time of the day, the postprandial glucose concentrations in the GIT lumen can vary in a large range and can be several times higher than in the blood. In rats fed with 65% glucose diet, the highest concentration of free glucose was found in the stomach (average about 1000 mmol/L during the day), while in the lumen of the proximal and distal small intestine it was about 50 and 1.0 mmol/L, respectively. In humans, concentration of gut luminal glucose about 48 mmol/L, was found in the upper intestine samples taken 2 h after eating a meal [2]. This shows that the gut provides both a barrier by limiting glucose passage and a regulatory mechanism for maintaining blood glucose levels. Thus, glucose absorption together with glucose ingestion and metabolism are all the interconnected processes determining the blood glucose levels and its availability to organs and tissues.

Disturbed regulation of the blood glucose levels leading to hyperglycemia, is a central problem of the pathophysiology of metabolic diseases such as obesity, metabolic syndrome and type 2 diabetes (T2D). Hyperphagia is a characteristic feature of these diseases suggesting a possible link to the regulation of blood glucose. In fact, the role of glucose as a satiety factor in the short-term regulation of appetite is well-known and was the basis for glucostatic theory of appetite [3]. Thus, molecular and cellular mechanisms underlying the dynamics of absorption of glucose (free or formed during the hydrolysis of complex carbohydrates) in the small intestine may impact on fluctuation of plasma glucose levels relevant to the regulation of appetite. However, it is not clear if targeting glucose absorption mechanism can be beneficial for treatment of hyperphagia in metabolic disease.

In recent years, various pathways of glucose transfer across the epithelium of the small intestine were studied concerning their contribution to the resultant absorption of glucose. Presently, most researchers agree that at relatively low glucose concentrations in the intestinal lumen in vivo (less than 30 mM), glucose (as well as galactose), is transferred across the apical membrane of enterocytes by active transport mediated by the transporter SGLT1, while its exit into the blood flow is carried out by the facilitated diffusion mediated by the transporter GLUT2, localized in the basolateral membrane [4,5,6].

At high luminal concentrations of glucose (more than 30 mM), the active transport of glucose becomes saturated and the other mechanisms might be involved in the absorption of glucose in the small intestine. It was hypothesized that one of these mechanisms may involve paracellular transfer through the tight junctions using a flow of absorbed water (‘solvent drag’ mechanism) [7]. Later, another hypothesis has been put forward that at high carbohydrate loads, the GLUT2 transporters can be quickly incorporated into the brush border membrane of enterocytes and participate in facilitated diffusion of glucose across this membrane [4]. The exact contribution of each of these mechanisms to the total absorption of glucose at its high concentrations in the intestinal lumen under normal conditions, as well as in metabolic disorders, needs further clarification.

Several studies have shown that glucose absorption as well as the expression and activity of the SGLT1 and GLUT2 glucose transporters in enterocytes are increased in diabetes [6], suggesting that it may contribute to hyperglycemia. Given the predominant role of SGLT1 transporters in the absorption of glucose in the small intestine, it is reasonable to consider these transporters as targets for the treatment of T2D. Indeed, recent use of new SGLT1 inhibitors was shown to reduce blood glucose levels and improve metabolic parameters in T2D patients without serious gastrointestinal side effects [8]. In the mouse, SGLT1 is predominantly expressed in the small intestine, with lower levels in the kidney, salivary glands and tongue; no expression in brain tissue was detected [9]. Nevertheless, despite a successful approach for lowering blood glucose levels, it is not clear if the inhibition of small intestinal glucose absorption may affect appetite and if can be useful for the treatment of hyperphagia.

The physiological studies showed that the expression and activity of the SGLT1 and GLUT2 transporters in small intestinal enterocytes undergo both short- and long-term regulation by dietary carbohydrates as well as by regulatory factors, including peptide hormones involved in the regulation of appetite such as leptin, glucagon-like peptide-1 (GLP-1) etc. [10]. However, the reported effects of such factors are often ambiguous, and in the case of GLUT2, they are even rare. Thus, in this paper we review the molecular mechanisms of glucose absorption in the small intestine in health and metabolic diseases and discuss their possible relevance to the regulation of appetite.

## 2. Mechanisms of Glucose Absorption in Normal Conditions

It is now generally accepted that in the range of relatively low concentrations of glucose (less than 30 mM) in the lumen of the small intestine, for example after eating a low-carbohydrate diet, the main route of absorption of this monosaccharide through the intestinal epithelium in vivo involves its active transport across the brush border membrane of enterocytes using the Na^+^-glucose co-transporter (SGLT1) [4,5,6] (Figure 1).

From the enterocytes, glucose is released through the basolateral membrane by facilitated diffusion with the participation of the GLUT2 transporter [4,5,24]. The absence of the SGLT1 transporter on the absorptive intestinal surface in humans with impaired glucose and galactose absorption is the most significant evidence of the need for this transport protein for the assimilation of these food components [25]. 

In the presence of high concentrations of glucose (more than 30 mM) in the lumen of the small intestine, arising after eating, active glucose transport is saturated and additional mechanisms may be involved in the absorption of glucose in the small intestine. Until now, the assumption that one of such mechanisms may be intercellular glucose transport in a stream of absorbed water has not lost its significance [7]. At the same time, in recent decades, studies on various animal species have shown that an additional mechanism that promotes the absorption of glucose in the small intestine in the range of its high concentrations in the intestinal lumen may also be facilitated diffusion across the brush border membrane of enterocytes with the participation of GLUT2 transporters, which are able to quickly integrate into this membrane [4,6].

### 2.1. Glucose Transporters SGLT1 and GLUT2

The SGLT1 transporter belongs to the family of sodium-glucose cotransporters SLCA5 and is expressed in significant amounts in the membrane of the brush border of small intestine enterocytes in humans and other animals [5]. The SGLT1 transporter, first cloned in 1987 in the rabbit, has a molecular weight of about 75 kDa [26,27]. The Kt for the transfer of glucose by this transporter, corrected for the effect of the preepithelial diffusion layer, in rat was 3–7 mM [28]. SGLT1 transports glucose and sodium ions in a 1:2 ratio. The driving force for this transport is created by the concentration gradient of Na^+^ ions across the brush border membrane of enterocytes, which is provided by Na^+^-K^+^-ATPase localized in the basolateral membrane and pumping Na^+^ ions out of cells. The substrate specificity of the apical glucose transporter in intestinal cells is the following: D-glucose > D-galactose > D-methylglucoside > D-3-O-methylglucose >> L-glucose, 2-deoxy-D-glucose [26].

The SGLT1 transporter can be competitively inhibited by phloridzin. The existence of a transport system on the basolateral membrane of enterocytes, which is inhibited by phloretin (phloridzin aglucon), was apparently first reported in 1976 by Kimmich and Randles [29]. Later, when studying the vesicles of the basolateral membrane of enterocytes, it was shown that, in contrast to the apical transport system, the basolateral transport system is characterized by a high Kt value (about 27 mM) for glucose and a wide range of specificity decreasing in the following order: 2-deoxy-D-glucose > D-glucose > D-galactose > D-3-O-methylglucose > D-mannose > D-xylose > D-fructose [30]. Later, when the GLUT2 transporter was isolated from the basolateral membranes of intestinal cells, it was shown to transport glucose, galactose, and fructose across the basolateral membrane in a Na^+^-independent manner [31]. It was also confirmed that the GLUT2 transporter has a low affinity for glucose (Km 20–40 mM), and showed its high affinity for D-glucosamine (Km 0.8 mM) [32]. In the early 2000s, it was shown that with a high carbohydrate load in the small intestine in different mammalian species, the GLUT2 transporter is localized in the brush border membrane of enterocytes, and therefore, together with the SGLT1 transporter, can participate in the absorption of glucose from the gut lumen [33].

### 2.2. Paracellular Glucose Transport

At the end of the last century, Pappenheimer proposed that in natural conditions the glucose absorption at high glucose concentrations in the intestine occurs mainly through its paracellular transfer on the flow of absorbed water [7]. This hypothesis was based on the significant discrepancy between the kinetics of glucose absorption in the small intestine previously observed in in vitro experiments and in acute in vivo experiments. Indeed, the results of in vitro studies have shown that in the range of high glucose concentrations (>30 mM), an almost complete saturation of the active transport by SGLT1 occurs, since the Michaelis constant for active Kt transport is 3–7 mM, corrected for the effect of the preepithelial diffusion layer [28]. However, in acute in vivo experiments, intestinal glucose absorption in the region of high glucose concentrations increased almost linearly with increasing substrate concentration in the intestinal lumen. This suggested that the unsaturated (passive, “diffusion”) component, apparently, significantly exceeds the maximum level of active transport mediated by SGLT1 [7].

The hypothesis of the paracellular glucose transport was consistent with the data that the absorption of glucose in the small intestine is accompanied by an increase in water absorption [34,35]. The author of the hypothesis also suggested that the triggering mechanism of paracellular transport is active glucose transport, creating an osmotic substrate gradient in the zone of tight intercellular contacts, ensuring fluid absorption. Indeed, an increase in the clearance of a number of lipophilic inert substances (from creatinine with a molecular weight of 113 D to inulin with a molecular weight of 3500 D) were found in the presence of luminal glucose (25 mM). In addition, it has also been shown that the glucose-induced increase in the permeability of the intercellular pathways in the intestinal epithelium is accompanied by structural changes in the tight intercellular contacts [36,37,38]. In particular, large dilatations within absorptive cell occluding junctions were revealed by electron microscopy. Solute-induced structural alterations were also associated with condensation of microfilaments in the zone of the perijunctional actomyosin ring, typical of enhanced ring tension [36,38].

The most weighty argument in favor of the paracellular glucose transport werevery high glucose concentrations in the lumen of the small intestine (up to 300–500 mM) that supposedly exist in humans and animals after eating [7]. However, these published data were obtained either under non-physiological conditions or by using insufficiently accurate glucose concentration assays in the chyme samples [2]. In the other study, specifically designed to determine the concentration of glucose in the intestinal lumen in several species (rats, rabbits and dogs) the concentration of luminal glucose during physiological digestion was found to be in the range of 10–30 mM and rarely exceeds 50 mM [2].

In our chronic experiments on rats at glucose concentrations in the lumen of the small intestine about 50–75 mM, the paracellular transport of glucose on the flow of absorbed water was evaluated to be less than 15% of the total absorption of glucose [39]. 

A predominant transfer of glucose through the intestinal epithelium in the water flow (“solvent entrainment”) under the conditions of real digestion has not been supported by other studies [40,41]. 

Nevertheless, it should be noted that paracellular transport of substances is characterized by significant species differences. For example, it has been shown that in bats, unlike flightless mammals, paracellular transport contributes to more than 70% of total glucose absorption. This is due to the difference in gut permeability with respect to the size of nutrient molecules [42]. Paracellular transport, apparently, is also one of the main mechanisms of glucose absorption in the intestine of birds [43]. Although the smaller intestine mass is compensated by its increased permeability, there is a lower selectivity of the system in comparison with the transport mechanism mediated by specific transporters. This leads to the fact that the toxins present in food can be absorbed from the intestinal lumen [44].

## 3. Adaptive Reactions of Transporters SGLT1 and GLUT2 in Physiological Conditions

There is a time scale for the adaptation mechanism of nutrient absorption and, accordingly, two types of reaction: rapid reactions, when the adaptation time is several minutes or hours, and slow reactions, when the adaptation time takes 1–3 days, which is comparable to the renewal time of the intestinal epithelium [45]. The mechanism of adaptation of nutrient absorption is aimed at responding to the prevalent type of nutrients, including carbohydrates, ingested at short and long terms. Peptide hormones produced in the gut and endocrine tissues are involved in the regulation of both rapid and slow reactions while gut bacteria may influence glucose absorption mainly via slow reaction. Below we discuss in more detail these regulatory processes.

### 3.1. Rapid Adaptive Responses to Glucose 

After a 30-min load of glucose (25 mM) into the lumen of the rat small intestine, both in vivo and in vitro, an increase in Vmax of active glucose transport with the participation of SGLT1 in the brush border vesicles of enterocytes was observed, while both the facilitated and passive components of the absorption of this monosaccharide did not change [46]. Using oocytes expressing the rabbit transporter SGLT1, it was observed that rapid changes in Vmax of active glucose transport, was accompanied by increased cell surface area and the number of SGLT1 transporters in the plasma membrane upon activation of protein kinase A (PKA) and protein kinase C (PKC) [25]. Moreover, the studies of the rapid regulation of human SGLT1 expressed in oocytes revealed that an increase in SGLT1 in the plasma membrane under the influence of glucose was due to an increase in exocytosis of these transporters from the Golgi apparatus, which was modulated by the protein RS1 [47]. A number of studies have noticed rapid changes in glucose transport through the basolateral membrane of enterocytes, caused by increased expression and activity of the GLUT2 transporter in this membrane under the influence of glucose infused in the intestinal lumen or after its intravenous administration [45].

In addition, a coordinated change in the activities of SGLT1 and GLUT2 and their mRNA in enterocytes of the small intestine in several mammalian species under the influence of a circadian rhythm has been described [48,49]. Indeed, the SGLT1 and GLUT2 activities were maximal with food intake, while the daily rhythm of SGLT1 remained even in an isolated segment of the jejunum [48]. These studies suggest that changes in the expression and activity of SGLT1 and GLUT2 in the small intestine are due to the appearance of nutrients in the intestine, and may also be regulated by biological clock genes [49].

In the early 2000s, Kellett’s group showed that with a high carbohydrate load (more than 30 mM) in the small intestine, GLUT2 transporters can quickly (within a few minutes) penetrate the membrane of the brush border of enterocytes and, as a result, appear to provide effective glucose absorption under these conditions [50,51]. Moreover, it has also been suggested that at high substrate concentrations in the small intestine, GLUT2-facilitated glucose diffusion becomes the main mechanism of glucose absorption, since active SGLT1-mediated glucose transport is almost completely saturated. It is assumed that the triggering mechanism of this process is the maximal active glucose transport in the intestine [4,42,43]. The sequence of events in this process was assumed as follows: the co-transport of Na^+^ and glucose by SGLT1 into the enterocyte induces depolarization of the brush border membrane, which activates the voltage-gated L-type calcium channel Cav1.3, which in turn, induces Ca^2+^ influx. Increase in intracellular Ca^2+^ results in remodeling of the terminal web and cytoskeletal structures necessary for the trafficking of GLUT2 to the apical membrane. In this manner, Ca^2+^ has the stimulating effect on the presence of PKCβII at the brush border membrane [51,52]. There is also evidence that an increase in the GLUT2 transporters in the enterocyte brush border membrane is due to increased exocytosis of GLUT2-containing vesicles [4].

Because translocation of GLUT2 toward and from the apical membranes occurs quite quickly (within 5–10 min) [51], the apical presence of GLUT2 has been long-time unnoticed. Indeed, under “normal” nutrition, a small and as yet poorly defined amount of GLUT2 is constitutively present in the apical membrane of enterocytes [53]. In another study conducted 15 min after mice were given a glucose bolus of 4 g/kg body weight, the authors could not find any evidence for the apical presence of GLUT2, but they did not rule out that a small amount of GLUT2 may be present in the brush border membrane of enterocytes, but its amount did not change after glucose loading in this membrane under experimental conditions [54]. Noticeable levels of apical GLUT2 were found at glucose concentrations above 30–40 mM, but much higher levels become apparent when the luminal glucose concentration exceeds 75 mM [52]. It is quite possible that the failure to determine the presence of the GLUT2 transporter in the brush border membrane of enterocytes in the case of a high concentration of glucose in the lumen of the small intestine in mice in the above study is associated with the authors’ use of experimental conditions to which the GLUT2 transporter is sensitive [54]. Immunohistochemical studies confirmed the incorporation of the GLUT2 transporter into the apical membrane of enterocytes during perfusion of an isolated segment of the jejunum of rats in chronic experiments (in the absence of anesthesia) with a high (75 mM) glucose solution [38,55]. In addition, in a mouse study, increased absorption of α-methyl-D-glucoside (AMG) and the content of the SGLT1 and GLUT2 proteins in the vesicles of the brush border of enterocytes were found in the small intestine 30 min after the glucose gavage. However, both with and without glucose loading, the capacity of active transport with SGLT1 was higher than that of facilitated diffusion with GLUT2 [56].

From a methodological point of view, it should be noted that the relative contribution of various mechanisms (active transport mediated by SGLT1 and passive diffusion mediated by GLUT2) to the total glucose absorption in the small intestine may significantly depend on the experimental in vivo approaches. For example, surgery and narcosis accompanying acute and chronic in vivo experiments significantly influence the levels and kinetics of nutrient hydrolysis and absorption in the small intestine [57,58,59]. In this regard, it is possible that different mechanisms of glucose absorption in the small intestine may be differently sensitive to anesthesia.

Moreover, a quantitative evaluation of the relative contribution of different mechanisms of glucose absorption to the total glucose absorption may also depend on the method of calculation of the corresponding kinetic constants; in particular, whether the effects of the pre-epithelial diffusion layer and the peculiarities of the intestinal absorptive surface (presence of villi) are included in the calculations [28,59,60]. For example, according to the theoretical estimates, the saturation of the SGLT1 starts on the top of the villi and then, with a further increase of the luminal substrate concentration, it continues along the lateral surface of the villi to the direction of the crypt [60,61]. Without considering the functional geometry of the intestinal surface, the estimates of Vmax of active glucose transport by SGLT1 in vivo can be underestimated, while the facilitated diffusion of GLUT2, on the contrary, is overestimated. This might be especially important in the case of diabetes, accompanied by hypertrophy of the intestinal mucosa [62].

### 3.2. Slow Adaptive Responses

A slow (within 1–3 days) increase in glucose absorption, SGLT1 and its mRNA content in enterocytes were observed in several mammalian species in response to an increase of dietary carbohydrates [63,64,65]. It is noteworthy that an increase in the mRNA of the SGLT1 transporter was observed when rats were fed diets containing various carbohydrates (glucose, galactose, fructose, mannose, xylose, or 3-O-methylglucose) [65]. This indicated that the marked increase in mRNA of the SGLT1 transporter does not depend on its carbohydrate metabolism in the body and whether it is transported with the participation of SGLT1. Moreover, in these experiments, the use of diets with high levels of glucose, galactose and fructose (but not AMG, mannose or xylose) increased the expression of the GLUT2 transporter mRNA in enterocytes. However, it remained unclear in which enterocyte membrane (basolateral membrane or brush border membrane) the GLUT2 protein content could be increased. Furthermore, in other experiments, it was shown that in high-glucose and high-fructose diets, the GLUT2 transporter is localized both in the basolateral and in the brush border membrane of the enterocytes [66]. In order to elucidate the response of the active and passive components of glucose absorption in the small intestine to a long-term increased luminal glucose load, we conducted chronic experiments on rats using long-term (1.5 h daily for 6 and 14 days) loads of an isolated segment of the small intestine with glucose solutions 25 mM (normal) and 125 mM (high) concentration. The results showed that an increase in glucose absorption in response to a high glucose load developed mainly due to the active component of glucose transport mediated by the SGLT1 transporter, but not due to the passive component evaluated using a mathematical approach [61].

It was shown that the adaptive response of glucose absorption by SGLT1 induced by an increase in carbohydrates in the diet (after 12 h), is first programmed in immature enterocytes localized in intestinal crypts. These enterocytes then migrate over the next 3 days to the tops of the villi, where they provide increased glucose absorption [67]. In mouse experiments, where the taste receptor or G protein α-gustin expressed in G cells, were removed, it was revealed that the long-term regulation of the SGLT1 protein and its messenger mRNA depends on the taste reception [68,69].

### 3.3. Responses to Peptide Hormones

Literature data show that insulin, glucagon, leptin as well as enteral hormones and neuronal activation may influence the intracellular trafficking and activity of SGLT1. We briefly review the effects of some of these messengers on glucose absorption bearing in mind that they are also involved in the regulation of appetite, mainly by stimulating satiety.

Two studies reported that at physiological concentrations, insulin increased glucose absorption in rat small intestinal preparations and during jejunal perfusion in anesthetized rats involving increased expression of SGLT1 [70,71]. In addition, insulin infused in the portal vein rapidly increased absorption of glucose in the small intestine. It has been suggested that this stimulation is mediated by the binding of insulin to receptors in the portal vein that activate the hepatic-intestinal nerves and increase SGLT1 in the brush border of enterocytes [72]. However, in other studies, after infusion of insulin (10 nM) into the jejunal vessels of rats, there was a decrease of glucose absorption from the jejunum, but an increase of glucose uptake from the vessels [73,74]. Moreover, insulin normalized the increased activity of SGLT1 in rats with streptozotocin-induced diabetes [75,76]. It was also reported that insulin can reduce transepithelial glucose transfer and GLUT2 protein expression on the brush border and basolateral membranes of Caco2/TC7 cells, and inhibits small intestinal fructose uptake in mice by decreasing the level of GLUT2 on the brush border membrane of enterocytes [77]. Thus, although the action of insulin on intestinal glucose absorption appears multidirectional, the underlying mechanisms are most likely post-transcriptional [78].

Incubation of enterocytes with glucagon (glucagon-29, that is released by the pancreas) for 15 min increased the uptake of galactose sensitive to phloridzin, which was accompanied by an increase in intracellular cAMP [79]. Moreover, glucagon stimulated glucose uptake from the rat jejunum by increasing the electrochemical gradient of Na^+^ ions [80]. Vascular infusion with glucagon-37 and glucagon-like peptide 2 (GLP-2), secreted by ileal L-cells, rapidly increased SGLT1 mediated glucose uptake in the small intestine [69,81]. The stimulation by glucagon-37, as in the case of glucagon-29, occurs after it binds to receptors on the basolateral membrane of enterocytes and the signal is cAMP- dependent [82]. The GLP-2 action on glucose absorption is likely to be carried out by an increased cAMP and activation of AMPK in enterocytes, which is a signal for the rapid acceleration of the SGLT1 exocytosis [83]. GLP-1 is also secreted by enteroendocrine L-cells and modulates postprandial glucose excursions mainly through potentiation of glucose-stimulated insulin secretion—i.e., the incretin function [84]. GLP-1 was shown to attenuate small intestinal glucose absorption in humans [85].

Gastric inhibitory polypeptide (GIP), also known as glucose-dependent insulinotropic polypeptide, is secreted by duodenal K-cells and has an inhibitory effect on histamine-induced gastric acid secretion [86]. There is evidence that GIP may stimulate intestinal glucose absorption in the jejunum by increasing, at least in part, its active transport mediated by SGLT1 [87].

Cholecystokinin (CCK) is a gastrointestinal peptide secreted rapidly by I-cells in the duodenum and proximal jejunum after food intake and is also abundantly produced in the brain. CCK is responsible for exocrine pancreatic secretion, gallbladder contraction, and intestinal motility [87]. It has been shown that CCK has a direct inhibitory effect on glucose uptake across the brush border membrane of enterocytes in rats, decreasing SGLT1 expression in this membrane [88].

Leptin is a protein hormone secreted mainly by fat cells, but was also found within the gastric mucosa [89]. A key physiological function of leptin is regulation of long-term energy balance and suppression of food intake [90]. In the small intestine, leptin binds to leptin receptors in the brush border membrane of enterocytes and inhibit SGLT1-mediated glucose transport, preventing PKC-dependent translocation of cytosolic SGLT1 transporters into the cell membrane [91,92]. Interestingly, oral administration of leptin has been reported to increase the level of GLUT2 mRNA in the jejunum, however, it is unclear if these changes may affect the absorption of glucose [93]. In addition, leptin may indirectly regulate the absorption of glucose in the small intestine by stimulating the secretion of CCK and GLP-1 [94,95].

### 3.4. Responses to Probiotics

Considering the differences in luminal and plasma concentration of glucose, it appears that a fraction of glucose in the gut is not absorbed and hence can be metabolized by gut bacteria. In fact, glucose is a preferred nutrient for growth of most bacteria. Gut bacterial composition is different in its energy extracting capacities from nutrients in obese and lean humans and, therefore, can influence luminal glucose levels and glucose absorption in the small intestine [96]. In this regard, several strains of mainly lactic acid bacteria (*Lactobacillus (L.) rhamnosus* GG ATCC53103, *L. acidophilus*, *L. casei* and *L. gasseri* BNR17) have been studied for their ability to suppress postprandial increases in blood glucose levels in normal conditions and in T2D [97,98,99]. The mechanism responsible for the lowering effect of lactic acid bacteria on postprandial blood glucose levels are not yet clear enough, and it was suggested that it may involve inhibition of intestinal α- and β-glucosidases [100].

It was also shown that the *Pediococcus pentosaceus* QU 19 strain isolated from Japanese fermented food rapidly lowered postprandial blood glucose levels in normal mice after a single administration. It turned out that this effect is associated with the assimilation of glucose by this strain in the intestinal lumen [101]. In a recent study, we found that *Enterococcus faecium* L3 reduced postprandial blood glucose and its absorption from the lumen of the small intestine as early as 3 days after starting daily administration of this probiotic strain to rats with experimental T2D (unpublished data).

Moreover, *Hafnia alvei* probiotic was recently shown to reduce food intake and glycemia in obese mice and overweight humans, potentially involving melanocortin-like effects of a bacterial protein caseinolytic protease B [102,103]. Effect of *Hafnia alvei* on glucose absorption has not yet been studied. Thus, since the role of gut bacteria in the regulation of appetite has been recognized, gut microbiota may provide a functional link between the regulation of glucose metabolism and appetite. For instance, mice lacking gut bacteria display increased expression of SGLT-1 and increased sucrose intake [104].

## 4. Adaptive Reactions of Transporters SGLT1, GLUT2 and Paracellular Transport in Metabolic Disease Conditions

### 4.1. Type 1 Diabetes Mellitus

Early studies using experimental models of type 1 diabetes in laboratory animals, for example induced by streptozotocin or alloxan, have reported increased glucose absorption as well as increase in small intestine mass and villous surface area [62,105]. In further studies in streptozotocin-induced diabetic rats an increased number of SGLT1 in enterocytes of the small intestine was demonstrated. Using autoradiography, it was revealed that additional glucose transporters in type 1 diabetes are localized mainly in the region of the middle and lower sections of the intestinal villi, and not in the area of the upper section of the villi, as in non-diabetic controls [11]. In other studies, rats with streptozotocin-induced diabetes displayed an increased SGLT1 mRNA levels as well as SGLT1-immunoreactivity and SGLT1-mediated transport in small intestine preparations and brush border membrane vesicles [12,13,14,15]. An increase in the level of the GLUT2 transporter in the basolateral membrane of enterocytes and its mRNA in the small intestine was also found in rats with streptozotocin-induced diabetes [12,13]. Furthermore, in some studies in rats with strozotocin-induced diabetes, increased expression of both SGLT1 and GLUT2 in the brush border membrane was noted [16,17].

Studies in people with type 1 diabetes and in animal models of type 1 diabetes have shown increased intestinal barrier permeability, which occurs before the onset of the disease [106,107]. In our recent study, we also observed an increase in the absorption of mannitol (a marker of passive intestinal permeability) in the small intestine in rats with experimental type 2 diabetes (unpublished data). We believe that the increased permeability of the intestinal barrier may augment the contribution of intercellular glucose transport in the overall absorption of this monosaccharide in the small intestine.

### 4.2. Type 2 Diabetes Mellitus

Similar to animal models of type 1 diabetes, a rat model of T2D (Otsuka Long-Evans Tokushima fatty rats) with impaired glucose tolerance before the onset of insulin resistance and hyperinsulinemia, displays increased expression of mRNA of the SGLT1 transporter [18]. However, leptin-deficient ob/ob mice exhibited an increased total intestinal glucose transport activity that was associated with increases in total intestinal dry weight and intestinal dry weight per centimeter, but not changes in glucose transport activity per unit of intestinal dry weight [108].

In our study, using rats with T2D caused by a high-fat diet and a low dose of streptozotocin, increased absorption of glucose in the small intestine was determined in vivo in the absence of anesthesia and surgery. There was also a tendency to an increase in the SGLT1 content and a noticeable decrease in the GLUT2 content in the apical membrane of enterocytes as determined by the immunohistochemistry, as well as a tendency to an increase in the number of enterocytes on the villi of the jejunum [20].

Increase of the SGLT1 protein and its mRNA in the brush border of enterocytes of the small intestine were found in patients with T2D [21,22]. Moreover, GLUT2 mRNA was increased in enterocytes in the duodenum of T2D patients, but no immunoreactivity in the brush border of these cells was detected. The latter may have been due to the fact that biopsy samples from the duodenum were taken after overnight fasting [21]. Nevertheless, another study reported that in patients with obesity and T2D, the content of GLUT2 was increased in the brush border and endosomal membranes, probably reflecting the resistance of enterocytes to insulin [23]. Thus, while the involvement of the SGLT1 transporter in increased glucose absorption through the apical membrane of enterocytes of the small intestine in both humans and experimental animals with T2D has been well established, the contribution of the GLUT2 transporter to this process needs further clarification.

### 4.3. Glucose Absorption as a Target in Diabetes Management

Based on the data revealing a key role of SGLT1 transporter in increased intestinal glucose absorption and resulting hyperglycemia in diabetes, a new approach to reduce postprandial hyperglycemia using SGLT1 inhibitors seems promising for the treatment of diabetic patients [8,109]. Nevertheless, clinical use of glucose transporter targeting drugs for treatment of diabetes is currently largely limited to SGLT2 inhibitors, for example gliflozins. SGLT2 is expressed in the kidney where gliflozins prevent glucose reabsorption leading to increased glucose loss in urine. Although most gliflozins have limited potency on SGLT1, they still may affect intestinal glucose absorption [110,111].

The experiments with SGLT1 inhibitors began with phloridzin, which has long been known to lower blood glucose levels. However, it has been abandoned as a potential type 2 diabetes drug due to its rapid hydrolysis to phloretin, which inhibits facilitative GLUTs transporters present in various tissues [112]. Recently, several selective SGLT1 inhibitors (KGA-2727, GSK-1614235, LX2761, JTT-662), as well as dual SGLT2/SGLT1 inhibitors (sotagliflozin, licogliflozin) have been tested in clinical trials showing reduction of blood glucose and improvement of metabolic parameters without serious gastrointestinal side effects [8,109,113,114]. The selective inhibitor SGLT1 (GSK-1614235) has shown that taking it before meals reduces the absorption of 3-O-methyl glucose in healthy people [113]. Dual inhibitor LX4211 reduced postprandial glucose, and increased GLP-1 and PYY levels in patients with T2D [114]. The stimulatory effects on GLP-1 were explained by the fact that inhibition of SGLT1 in the upper part of the small intestine may decrease glucose absorption and, thereby, promotes glucose delivery to the lower parts of the intestine, where it can be metabolized by microbiota which, in turn, can stimulate GLP-1 secretion by increased production of short-chain fatty acids. Having discussed this specific topic, we should not forget to mention that reducing carbohydrate intake should be the first choice and the safest way for lowering glucose transport from the gut to the blood.

## 5. Glucose Absorption and Appetite Regulation

Glucose absorption can be considered an integral part of the homeostatic system maintaining blood glucose levels, where it may provide the negative feedback to the brain control of food intake. Nevertheless, it should be noted that such homeostatic system can be overrun by the hedonic regulation of appetite, whereas highly pleasurable glucose intake my trigger new intakes via activation of the brain dopamine [115]. The relative changes or fluctuations of blood glucose levels as determined by simultaneous processes of glucose supply and utilization served as the basis for the “glucostatic” theory of appetite proposed by Jean Mayer in the 1950s [3]. In this section, we will discuss the possible relevance of small intestinal glucose absorption to the control of appetite and feeding behavior i.e., the physiological functions necessary for nutrient intake and glucose metabolism in both healthy and disease conditions (Figure 2).

It should be first recalled that appetite is a mostly visceral feeling of hunger and satiety which drive feeding behavior. The normal rhythms of appetite in humans, related to its short-term control, include about 20 min of satiation and 5 h of satiety terminating in a feeling of hunger which will persist until the next meal (Figure 2). In 2000 Schwartz and colleagues presented the energy homeostasis model of food intake control involving long-term metabolic signals derived from fat storage (leptin, insulin) coordinated with the short-term regulation of appetite, mainly involving satiety signals produced by the gastrointestinal tract [116]. Indeed, the secretion of intestinal peptide hormones is increased after a meal and along with the vagal efferents activate satiety.

Glucose may also play the role of a short-term satiety signal which plasma levels rise after each meal and returns to the preprandial levels after 1–2 h [117] (Figure 2). Increased plasma levels of glucose and several gut peptides activate neuronal satiety pathways and contribute to satiety mechanisms by inhibiting gastric emptying [118]. Some data, however, suggest that postprandial levels of plasma insulin rather than of glucose are associated with satiety [119]. Importantly, the different dynamics of postprandial glucose peak and glucose absorption do not support the latter to play a role in satiety signaling. In fact, in the postprandial period, glucose absorptions in the intestine begin after the gastric emptying which occurs 1–2 h after a meal, i.e., when the organism has been in the satiety state (Figure 2). The meal-induced peak of plasma glucose appears, hence, as an absorption-independent result of a neuronal reflex mechanism to nutrient ingestion involving mainly acute activation of hepatic glucose production [120]. The physiological significance of such mechanisms may be the anticipation of carbohydrate absorption extracted from nutrients and their storage as glycogen in the liver [121].

Duration of satiety of carbohydrate-rich meal is shorter than that induced by other macronutrients suggesting that postprandial changes of blood glucose, i.e., the rate of glucose absorption, may potentially influence regulation of hunger [122]. A special situation may exist involving dietary non-digestible fibers metabolized by gut bacteria resulting in production of SCFA which are known to promote satiety [123]. However, the satiety-increasing effect of fiber-rich diets in T2D subjects is inconclusive and may depend on the individual composition of gut microbiota [124].

The feeling of hunger occurs 5–6 h after a meal, corresponding to the average transit time in the small intestine, i.e., it occurs during the late phase of nutrients absorption. Early work by Anton Julius Carlson suggested that falls in blood glucose levels below post-absorptive levels may cause hunger, for example by inducing stomach hunger contractions [125]. However, such a possibility was not experimentally confirmed. Nevertheless, the observation in 1980 by Jeanine Louis-Sylvestre and Jacques Le Magnen of a brief decline in plasma glucose concentration by 6–8% 5 min before each meal in rats, raised a new possibility of the role of plasma glucose changes in meal onset [126]. Indeed, pre-prandial transient decline in blood glucose was also found in humans and it was associated with food requests [127]. In their in-depth review of this topic, Campfield and Smith concluded that a transient decline in blood glucose may represent an “endogenous signal” for meal initiation [128]. Le Magnen thought that short blood glucose declines by 8% before a meal was due to a decreased absorption of glucose in the intestine. However, the follow-up studies were not favorable to such conclusion and proposed other underlying mechanisms of such a brief and transitory decline of blood glucose, although its exact reason remains unknown [129].

Thus, intestinal glucose absorption does not seem to play a direct role in the short-term regulation of appetite, i.e., either in meal termination or its onset. It does not, however, exclude an indirect effect of the rate of glucose absorption in appetite control as a part of the long-term regulation of energy metabolism, for example, by contributing to the mechanism of hyperglycemia in T2D and obesity. Indeed, delayed satiety is a typical feature of obese humans and rodents involving long-term modifications of the brain circuitries regulating appetite, for example, the dopamine system [130]. Independent regulation of appetite and glucose absorption questions the rationale of glucose transporter inhibitors for the treatment of hyperphagia in T2D and obesity. In fact, despite hyperglycemic levels in the fasting state, diabetic and obese patients still display acute glucose peaks after a meal, a phenomenon already noticed by Jean Mayer [3]. A postprandial rise of glucose in T2D patients may be less efficient to activate satiety signaling via insulin secretion, due to insulin resistance [119]. Whether a mechanistic link may exist between increased intestinal glucose absorption and insulin resistance in T2D is presently unclear.

Nevertheless, lowering glucose absorption may create deficit in this macronutrient and trigger homeostatic mechanisms of hunger. For instance, inhibition of fat absorption in the gut by orlistat produced an anti-obesity effect but increased food intake [131]. Use of gliflozin in humans was also accompanied by an increased feeling of hunger and sugar intake [132,133]. One of the arguments for the potential utility of SGLT1 inhibition as a treatment of hyperphagia, assumes that prolonged elevated luminal levels of glucose may directly or indirectly (for example, via microbiota) activate secretion of intestinal satiety hormones, for example, PYY [134]. However, experimental data in rats and humans did not fully support this idea. As such, inhibition of SGLT1 in rats lowered plasma glucose insulin but had no effect on food intake and also reduced postprandial levels of satiety hormones GIP and PYY, and had no effect on GLP-1. In humans, premeal administration of a SGLT1 antagonist reduced both glucose and insulin but also GIP. Although increased levels of GLP-1 were found, no effects on food intake or appetite of the SGLT1 inhibition were reported [113].

## 6. Conclusions

Active transport of glucose mediated by SGLT1 in the apical membrane of enterocytes appears as the main molecular mechanism of glucose absorption in the small intestine. This mechanism determines the rate of glucose entry into the bloodstream under both low and high carbohydrate load in the gut. During high carbohydrate loading, paracellular transport with water flow and facilitated diffusion mediated by GLUT2 into the apical membrane of enterocytes further contribute to the total uptake of glucose.

In metabolic disorders characterized by hyperglycemia, such as in both types 1 and 2 diabetes, increased expression and activity of SGLT1 contributes to increased glucose absorption in the small intestine and, therefore, represents a pathophysiological factor of hyperglycemia. Reducing SGLT1-mediated transport of glucose appears, hence, as a valuable therapeutic target for diabetes treatment. Nevertheless, in spite of obvious metabolic advantages of the treatment of hyperglycemia, inhibition of small intestinal absorption of glucose does not appear as a valid target for hyperphagia. In fact, the plasmatic variations of glucose levels, associated with changes of appetite, occur independently from intestinal glucose absorption. Lowering plasmatic glucose levels may, hence, trigger unwanted effects of increased hunger by affecting long-term homeostatic regulation of energy metabolism. This suggests that the treatment of hyperglycemia based on SGLT1-mediated inhibition of small intestinal glucose absorption should be combined with other therapeutic strategies selectively targeting appetite control. Recent results of a clinical trial using *Hafnia alvei* in overweight subjects [103], suggest that development of new probiotics influencing both glycemia and appetite may represent a promising supplementary therapy for the pharmacological treatment of metabolic diseases.

## Figures and Tables

**Figure 1 nutrients-13-02474-f001:**
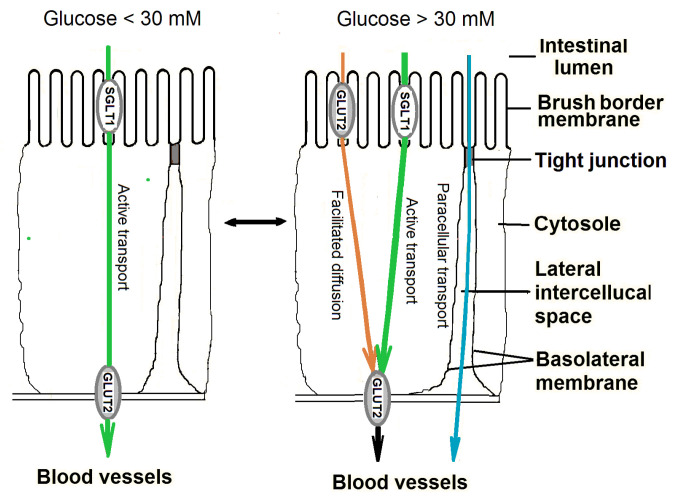
Molecular pathways of glucose absorption in the small intestine. Preferential localizations of SGLT1 and GLUT2 transporters in the brush border and the basolateral membranes of enterocytes determine the rate of glucose absorption under low (<30 mM) and high (>30 mM) luminal glucose concentrations in healthy conditions. In both ranges, changes in glucose absorption mechanisms are part of both fast and slow adaptive responses to dietary carbohydrates. These mechanisms are also under the modulatory effects of intestinal peptide hormones and potentially probiotics (see below). The same glucose absorption pathways and localization of SGLT1 and GLUT2 transporters are present in type 1 and type 2 diabetes, but in latter case, SGLT1-mediated glucose uptake is increased [11,12,13,14,15,16,17,18,19,20,21,22], and paracellular glucose transport in the water flux also appears to increase (see below). As for the facilitated diffusion of glucose through the apical membrane by GLUT2, the data are inconsistent [16,17,20,21,23].

**Figure 2 nutrients-13-02474-f002:**
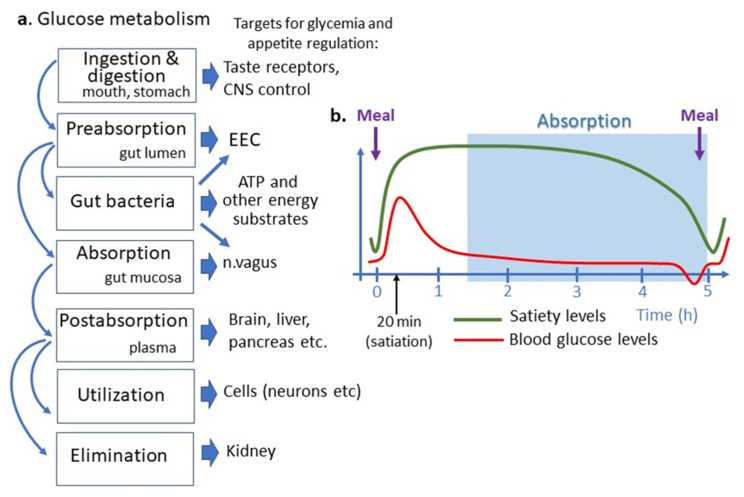
Place of intestinal glucose absorption in glucose metabolisms and regulation of appetite. (**a**) Different steps of glucose metabolism correspond to different targets for appetite regulation. (**b**) Temporal relation between appetite cycles shown as satiety levels, blood glucose levels and an approximate period of intestinal nutrient absorption. Note that the small intestinal glucose absorption occurs after gastric emptying and appearance of satiation i.e., later than postprandial glucose peak, which is due to reflectory hepatic glucose production. Intestinal glucose absorption continues during a short preprandial fall in blood glucose and appearance of hunger feeling. ATP, adenosine triphosphate, CNS, central nervous system, EEC, enteroendocrine cells.

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
