# Peer review of "Mechanisms of Glucose Absorption in the Small Intestine in Health and Metabolic Diseases and Their Role in Appetite Regulation"

_nutrients, 2021, doi:10.3390/nu13072474_

Round 1

Reviewer 1 Report

The present review written by Gromova and coll. gives a brief, comprehensive and pertinent overview of the current knowledge on the mechanisms of glucose absorption in the small intestine in physiological conditions and their alterations in metabolic diseases as well as their relevance to the regulation of appetite. The paper is very interesting, well written although the authors should be careful with the typing mistakes and the English language that needs to be improved sometimes. This review is logically organized and the reference list is complete and up-dated.

However, some points deserve clarification:

- In the introduction section, the authors should briefly describe the tissue distribution of SGLT1 in the organism (notably in the brain).

- lines 185 to 187: “glucose-induced increase in the permeability of the intercellular pathways in the intestinal epithelium is accompanied by structural changes in the tight intercellular contacts”; the authors should briefly develop. What are these structural changes?

- lines 189 to 211: The authors should shorten substantially these two § that is just a discussion of a paper that they consider doubtful.

- lines 253 to 267: Are the molecular mechanisms underlying the increased expression of SGLT1 and GLUT2 following an oral glucose load known. If yes, the authors should mention them.

- lines 360 to 380: The effect of insulin on SGLT1 and GLUT2 expression is not clear and seems controversial….Is it possible to clarify these effects of insulin?

- lines 395 to 399: GIP stimulates intestinal glucose absorption, but GIP is a well known incretin that dercreases plasma glucose levels….This is paradoxical. Could you comment on that?

- lines 418 to 444: this § on the impact of gut microbiota on glucose absorption is poorly documented. Is this chapter essential in the context of this review?

-  In the chapters 4.1 and 4.2, data with molecular mechanisms are missing. Please, briefly add them if available in the literature.

Reviewer 2 Report

Please see the attached document for my comments.  

Reviewer 3 Report

the topic is very interesting and the article is well written. Development of new probiotcis influencing both glycemia and appetite appear as a promising therapy for a supplementary pharmacological treatment of metabolic disease

Round 2

Reviewer 2 Report

The authors have well addressed my comments. There are two or three minor suggestions that the authors may want to consider.
